# Inequality of gender, age and disabilities due to leprosy and trends in a hyperendemic metropolis: Evidence from an eleven-year time series study in Central-West Brazil

José Francisco Martoreli Júnior[1]*, Antônio Carlos Vieira Ramos[1], Josilene Dalia Alves[2], Juliane de Almeida Crispim[1], Luana Seles Alves[1], Thaís Zamboni Berra[1], Tatiana Pestana Barbosa[1], Fernanda Bruzadelli Paulino da Costa[1], Yan Mathias Alves[1], Márcio Souza dos Santos[1], Dulce Gomes[3], Mellina Yamamura[4], Ione Carvalho Pinto[1], Miguel Angel Fuentealba-Torres[5], Carla Nunes[6], Flavia Meneguetti Pieri[7], Marcos Augusto Moraes Arcoverde[8], Felipe Lima dos Santos[1], Ricardo Alexandre Arcêncio[1]

**1** Department of Maternal-Infant Nursing and Public Health, University of São Paulo at Ribeirão Preto College of Nursing, Ribeirão Preto, São Paulo, Brazil, **2** Departament of Epidemiology, Federal University of Mato Grosso, Cuiabá, Mato Grosso, Brazil, **3** Department of Mathematics, University of Évora, Évora, Portugal, **4** Departament of Nursing, Federal University of São Carlos, São Carlos, São Paulo, Brazil, **5** Faculty of Nursing and Obstetrics, University of los Andes, Santiago, Chile, **6** Department of Public Health, New University of Lisbon, Lisbon, Portugal, **7** Department of Nursing, Londrina State University, Londrina, Paraná, Brazil, **8** Center for Education, Letters and Health, Western Paraná State University, Campus Foz do Iguaçu, Foz do Iguaçu, Paraná, Brazil

* jose.martoreli@usp.br

**Data Availability Statement:** The data from this manuscript entitled "Inequality of gender, age and

## Abstract

The present study aimed to investigate the epidemiological situation of leprosy (Hansen's Disease), in a hyperendemic metropolis in the Central-West region of Brazil. We studied trends over eleven years, both in the detection of the disease and in disabilities, analyzing disparities and/or differences regarding gender and age. This is an ecological time series study conducted in Cuiabá, capital of the state of Mato Grosso. The population consisted of patients diagnosed with leprosy between the years 2008 and 2018. The time series of leprosy cases was used, stratifying it according to gender (male and female), disability grade (G0D, G1D, G2D, and not evaluated) and age. The calendar adjustment technique was applied. For modeling the trends, the Seasonal-Trend decomposition procedure based on Loess (STL) was used. We identified 9.739 diagnosed cases, in which 58.37% were male and 87.55% aged between 15 and 59 years. Regarding detection according to gender, there was a decrease among women and an increase in men. The study shows an increasing trend in disabilities in both genders, which may be related to the delay in diagnosis. There was also an increasing number of cases that were not assessed for disability at the time of diagnosis, which denotes the quality of the services.

disabilities due to leprosy and trends in a hyperendemic metropolis: Evidence from an eleven-year time series study in Central-West Brazil" are available on Notifiable Diseases Information System (SINAN). After approval of the Institutional Review Board at the University of São Paulo, College of Nursing (EERP/USP) under Certificate of Presentation for Ethical Appreciation (CAAE)No. 0394720.3.0000.5393, data were provided by health surveillance service from the regional management unit, state government of Mato Grosso, on March 2020.The study required the approval of the Institutional Review Board at the University of São Paulo because the information collected could identify the participants. The variables adopted in this study include date when leprosy cases were reported in the SINAN (notification date), age and sex; we do not use any variable that can identify the participants. In accordance with the recommendations of PLOS ONE, we are providing the minimum set of anonymized data necessary to replicate our findings as Supporting Information files (S1_Dataset). The e-mail contact were we've managed to gather all the data used in this article is: sms.educacaopermanente@cuiaba.mt.gov.br Corresponding to the health authority (Secretaria municipal da saúde de Cuiabá) of the municipality.

**Funding:** This study was financed in part by the Coordenação de Aperfeiçoamento de Pessoal de Nível Superior - Brasil (CAPES) - Finance Code 001 (Process 304483/2018-4) (https://www.gov.br/capes/pt-br) to RAA. The funders had no role in study design, data collection and analysis, decision to publish, or preparation of the manuscript.

**Competing interests:** The authors have declared that no competing interests exist.

## Author summary

In the 2019 report, Brazil had a detection rate of 13.23 per 100.000 inhabitants far from the goal of less than 1 leprosy (Hansen's Disease) case per 10,000 inhabitants describe by the World Health Organization. The present study aimed to investigate the epidemiological situation of leprosy and its trend between 2008 and 2018 in a hyperendemic metropolis in the Central-West region of Brazil. A total of 9.739 leprosy cases were reported between 2008 and 2018. The majority of cases were male (58.37%), with a predominant age of 15 to 59 years (87.55%). The predominant level of education was incomplete elementary school (43.96%). The disability grade at diagnosis showed that 40.19% had G0D and for the G2D was 8,.06%.There was a predominance in operational classification of multibacillary cases (72.85%). While detection rate trends in females and the majority of the age groups are decreasing, increases are seen in the detection of male patients and patients already suffering from disabilities. Although declining trends were presented, the metropolis is still not close to elimination showing the need prioritize leprosy actions and to improve care for this disease.

## Introduction

Leprosy, also called Hansen's Disease, is a chronic infectious disease caused by *Mycobacterium leprae*, which affects Schwann cells, causing their destruction, affecting the skin, and resulting in severe neuropathies, which can lead to physical disabilities [1]. In the past 30 years, the World Health Organization (WHO) has sought measures to eliminate leprosy, and although its indicators have been decreasing over the years, the goal of elimination (prevalence <1 case per 10.000 inhabitants) has not yet been reached and currently seems to be more distant than imagined [2,3].

According to WHO 2020 Weekly epidemiological record Global leprosy, India, Brazil and Indonesia reported >10,000 new cases in 2019, classifying them as the three most highly endemic countries [4]. In 2019, Brazil had a detection rate of 13.23 per 100.000 inhabitants [5,6].

WHO stated in the "Towards zero leprosy Global leprosy (Hansen's Disease) strategy 2021–2030" [1], that the main actions to control the disease should be directed towards leprosy prevention upscaling alongside integrated active case detection, leprosy disease management and its complications and prevent new disability and combat stigma and ensure human rights are respected. Interruption of transmission and elimination of disease are at the core of their strategy document, as well as of the WHO "Guidelines for the Diagnosis, Treatment and Prevention of Leprosy" [1,7].

Brazil is a country of continental proportions, which makes leprosy control a major challenge. It is divided into five macro-regions, of which the Central-West is one of the most problematic regions in terms of the burden of the disease. A study carried out in that region showed that in the trienniums of 2001–2003 and 2010–2012, there was a reduction in the disease; however, there are geographical areas where leprosy control has not advanced and these locations are far from elimination [8].

The official reports from Brazil indicate the reduction of leprosy in general, but there is evidence that there is gender inequity in access to health services [9]. Another issue refers to age, as there is evidence that the population aging process is changing the profile of the leprosy morbidity profile, given the number of people falling ill in a context of poverty and inequality, with older adults having more difficulty in accessing health services and tending to have a

more unfavorable prognosis [10]. This needs to be better addressed from the perspective of health surveillance.

Also, regarding the inequality related to age, it is known that when there is a delay in diagnosis, children who had contact with index cases also become ill, which is an important gap to be filled [11]. Accordingly, the elimination of leprosy involves comprehending the determinants, according to a gender and age equity perspective. It is also understood that gender and age inequity should not be analyzed only from the perspective of detection, but also in terms of disability, as the WHO recognizes the disability grade indicator as the most sensitive measure of the real leprosy situation in a community [1].

There are hypotheses that there are gender differences, and that older adults and children are more severely affected by the disease, with regard to disabilities [10,11]. These aspects need to be studied in order to define public health policies and plan strategic actions in priority areas, as well as to advance equity. There are several tools that could be used to test these hypotheses; among them, one of the more sensitive tools is the time series. Its use is justified in the field of public health, as it can show the trend of the disease in vulnerable groups and verify how much success has been achieved in terms of the goal of elimination and reduction of injustices and/or inequity [12].

This study aimed to investigate the epidemiological situation of leprosy and trends in the detection of cases and disabilities, and to evidence disparities and/or differences regarding gender and age in a hyperendemic metropolis in Central-West Brazil.

## Materials and methods

### Ethics statement

The study was approved by the Ethics Committee of the University of São Paulo, School of Nursing (EERP/USP) under CAAE: 30394720.3.0000.5393. The investigation was exempted from signing consent forms, as it used secondary data, considering that the data were analyzed in an aggregated manner, without individual identification. Anonymized data was sent as Supporting Information to the Journal available to ensure reproducibility.

### Study design and setting

This ecological time series study was carried out in Cuiabá [13], capital of the state of Mato Grosso, located in the Central-West region of Brazil (Fig 1). The metropolis has an area of 3.266,538 km$^2$, with an estimated population of 607.153 inhabitants and demographic density of around 185.87 inhabitants per km$^2$ in 2018 [14].

In relation to socioeconomic indicators, Cuiabá has an illiteracy rate of 4.56% for women and 4.79% for men, a life expectancy at birth of 75 years of age, and a Human Development Index (HDI) of 0.785. It also has a Gini index of 0.59, an indicator that measures how equitably a resource is distributed in a population; the nearest to 0 is the population with lowest inequality, and closest to 1 is the most unequal [15].

Regarding basic sanitation, 53.52% of Cuiabá's territory has a sewage network and 98.12% a water supply [16,17]. It also has 63 primary health units, distributed throughout 4 administrative regions North, South, East and West.[18,19]. It should be highlighted that the municipality provides the following referral services for procedures of greater technological complexity as well as for leprosy,: "CERMAC—*Centro Estadual Regional de Média e Alta Complexidade*" (dermatology surveillance services), "*Hospital Metropolitano*" (hospital care for leprosy surgery services), "*Hospital Universitário Júlio Muller*" (hospital care for ophthalmology referral), "CRIDAC CER III—*Centro de Reabilitação Integral Dom Aquino Corrêa*" (center specialized in rehabilitation and regional outpatient and hospital referrals) [20,21].

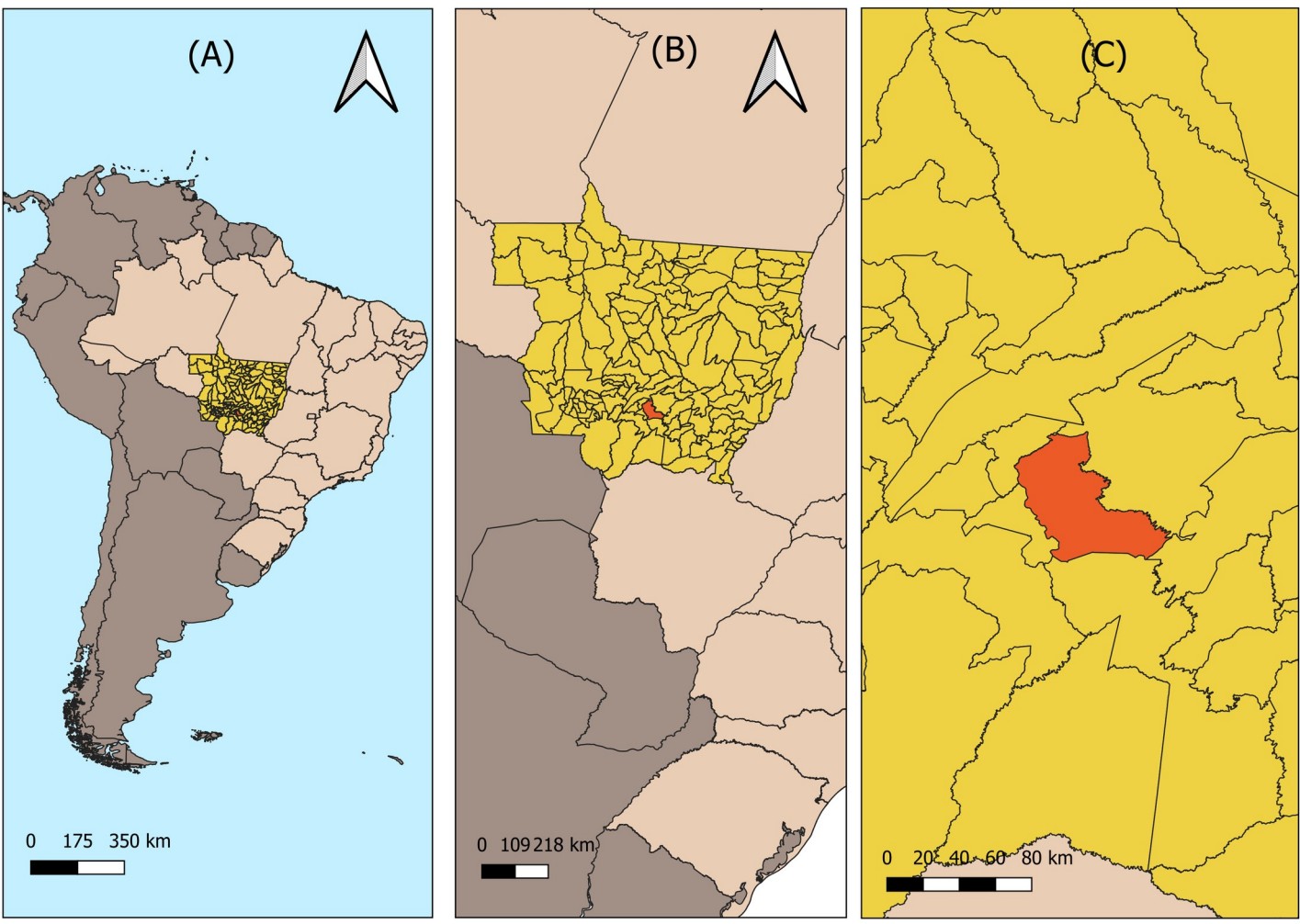

**Fig 1. Location of Cuiabá—Mato Grosso.** (A) Brazil; (B) State of Mato Grosso; (C) City of Cuiabá. Source: Instituto Brasileiro de Geografia e Estatística (IBGE)–All maps are in public domain. (https://portaldemapas.ibge.gov.br/)

The municipality presents a detection rate of 45.30 cases for every 100.000 inhabitants, classifying it as hyperendemic for leprosy according to "Sistema de Informação de Agravos de Notificação–SINAN" (Notifiable Disease Information System) 2018, which classifies hyperendemic cities with a detection greater than 40 cases for every 100.000 people [22,23]. In children under 15 years of age, in 2018 the detection rate was 7.9/100.000 inhabitants, placing the metropolis in the very high category for children [22].

## Study population and information sources

Leprosy cases registered in the SINAN from 2008 to 2018 of residents of the city of Cuiabá were included. The SINAN is the Brazilian information system responsible for recording and processing information on mandatory notifiable diseases such as leprosy throughout Brazil, providing bulletins and reports of morbidity and constituting one of the main surveillance systems in the country.

According to the Brazil Practice Guide for Leprosy, the cases are diagnosed if the person presents a defined skin area with altered or complete loss of sensitivity with or without

compromised nerve, or nerves with neural thickening, and/or positive bacterial index (via skin smear). The bacterial index is not the defining factor but important for clinical and epidemiological evaluation [24].

The selected variables were date of notification of the case, gender (male, female), age, education (no schooling, incomplete elementary education, complete elementary education, incomplete high school education, complete high school education, incomplete higher education and complete higher education), WHO operational classification (paucibacillary [pb], multibacillary [mb]), clinical form based on Madrid classification (indeterminate, tuberculoid, borderline and lepromatous) and assessment of disability grade in the diagnosis (Grade 0 disability [G0D], Grade 1 disability [G1D], Grade 2 disability [G2D], and not evaluated) [25]. Below, in the Table 1 is shown the disability grade characteristics.

Access to the SINAN database was obtained from the Health Surveillance Service of the Regional Health Management of Cuiabá ["*Serviço de Vigilância Sanitária da Gerência Regional de Saúde de Cuiabá*"] in November 2019.

## Statistical analysis

Initially, the variables of interest were standardized, with data relating to age, gender, and education considered in the sociodemographic dimension and operational classification, clinical form, and disability grade at diagnosis in the clinical-epidemiological dimension. Descriptive analysis of the sociodemographic and clinical-epidemiological variables was performed.

Next, the time series of leprosy cases were constructed according to the total number of cases [26,27], gender (male and female), and disability grade (G0D, G1D, G2D, and not evaluated).

For the construction of the time series, the general rates of detection and those stratified by gender were calculated, considering the total population of the municipality (for general detection rate) and the populations of men and women (for rates stratified according to gender) as the denominator, all with a multiplication factor per 100,.000 inhabitants.

After the construction of the time series, according to detection in general by gender, age, and disability grade, the evolutions of the trends were calculated using the Seasonal-Trend decomposition procedure based on Loess (STL) [28]. This methodology is based on the classic decomposition of time series that disaggregates the total series into three additive components (trend, seasonality or error), allowing each of these components to be separately estimated and identifying the source of variability of the series in a more concise way than through a global

**Table 1. Disability grade and his characteristics.**

| Grade | Characteristics |
|---|---|
| 0 [G0D] | No problem with eyes, hands or feet's due to leprosy |
| 1 [G1D] | Reduction or loss of eye sensitivity<br>Reduction or loss of sensation in hands and/or feet (does not feel 2g or pen touch) |
| 2 [G2D] | Eyes: lagophthalmos and/or ectropion; trichiasis, central corneal opacity; acuity visual less than 0.1 or not counting fingers at 6m.<br>Hands: trophic injuries and/or traumatic injuries, claws; resorption, hand down<br>Feet: trophic and/or traumatic injuries, claw hand deformity, resorption, foot dropped, contracture of ankle |

Source: Ministério da saúde, Guia prático sobre a Hanseníase–Secretaria de Vigilância em Saúde–Departamento de Vigilância e Doenças Transmissíveis—Brasília—DF 2017 [24], based on WHO disability grading for leprosy (https://apps.who.int/iris/handle/10665/42060).

analysis of the series. The STL has a simple design that consists of a sequence of applications of the Loess, allowing analysis of the properties of the procedure and quick calculations [29].

One of the advantages of this methodology is that it is quite robust regarding the existence of outliers. 'Trend' refers to the general direction in which the variables of the time series develop, according to a time interval, presenting a pattern of increase/decrease of the variable over a certain period. 'Seasonality' is reflected in identical patterns that a time series seems follow and that occur regularly at fixed periods of time. Finally, 'noise' is the fluctuations that occur over the time of the series, visualized as irregular and random movements perceptible only with the removal of the other components [30].

Having estimated the three components of the time series, only the trend was selected to characterize the trend of the variables of interest over time. Subsequently, the Average Monthly Percentage Change (AMPC) was calculated for the trends in the general detection rates by gender and cases with disability grade identifying the mean percentages and how much the trends increased or decreased over the study period. Finally, the trends of the disability rates in various groups were also analyzed. All analyses were performed using the R Studio version 3.5.2 statistical software.

## Results

A total of 9,739 leprosy cases were reported between 2008 and 2018. As shown in Table 2, the majority of cases were male (58.37%), with a predominant age of 15 to 59 years (87.55%). The predominant level of education was incomplete elementary school (43.96%).

Regarding the clinical variables, there was predominance in operational classification of multibacillary cases (72.85%), and for the clinical form by borderline cases (56.83%). The disability grade at diagnosis showed that 40.19% had G0D, followed by 30.45% that were not evaluated, 21.3% with G1D and 8.06% with G2D.

According to the results, for the detection rates (S1 Fig) there was a decreasing trend in the general detection rate and for the reported female cases, and an increasing trend for male cases (0.01%).

Fig 2 presents the main changes in the structure of the time series of the total detection rate of leprosy cases. Three changes of structure are verified in the series; the first occurred in November 2011, in which the series shows a marked decrease until the year 2013. The second change in structure occurred in August 2013, with an increase in the time series, presenting the highest values in the years 2014 and 2015, with a detection rate of more than 20 cases per 100.000 inhabitants. From October 2015 onwards, the last change in structure occurred, with a decrease and later stability in the detection rate values until the end of the series.

Considering disability grade in the general population (S2 Fig), no disability (G0D) was the only grade with a decreasing trend, while G1D (0.56%), G2D (0.38%), and unevaluated cases (0.28%) showed increasing tendencies.

The increasing or decreasing of the trends is shown by the Table 3 below.

When stratifying disability grade according to gender (S3 Fig) there was a decreasing trend for no disability (G0D) in males (-0.44%) and females (-0.70%), and a decrease in G2D in females (-6.09%). The other disability grade showed increasing trends throughout the series.

In S4 Fig shows the disability grade according to age groups. For all age groups, the rates of new diagnosis with no disability (G0D) are decreasing. Regarding G1D, all age groups showed an increasing trend. For G2D, only the 30–59 years age group presented a decreasing trend (-0.62), with the other groups showing increasing trends. Finally, regarding those not evaluated for disability grade, children aged under 15 years (-0.42) and the group aged 15 to 29

**Table 2. Clinical and social epidemiological characteristics of the cases diagnosed with leprosy, in an endemic municipality in Central-West Brazil (2008–2018).**

| Variables | Frequency ($n$ = 9739) | % |
|---|---|---|
| **Gender** | | |
| Male | 5685 | 58.37 |
| Female | 4052 | 41.60 |
| Not classified/incomplete | 2 | 0.03 |
| **Age** | | |
| <15 years | 514 | 5.28 |
| 15 to 59 years | 8527 | 87.55 |
| 60 years or more | 698 | 7.17 |
| **Education** | | |
| No schooling | 738 | 7.58 |
| Incomplete elementary education | 4279 | 43.96 |
| Complete elementary education | 821 | 8.43 |
| Incomplete High School Education | 736 | 7.55 |
| Complete high school education | 1621 | 16.64 |
| Incomplete higher education | 262 | 2.70 |
| Complete higher education | 552 | 5.65 |
| Not classified/incomplete | 650 | 6.67 |
| Not applicable | 80 | 0.82 |
| **Operational classification** | | |
| Paucibacillary | 2582 | 26.51 |
| Multibacillary | 7095 | 72.85 |
| Not classified/incomplete | 62 | 0.64 |
| **Clinical form** | | |
| Indeterminate | 1118 | 11,48 |
| Tuberculoid | 1456 | 14,95 |
| Borderline | 5535 | 56,83 |
| Lepromatous | 1422 | 14,60 |
| Not classified/incomplete | 208 | 2,14 |
| **Disability grade at diagnosis** | | |
| Grade 0 | 3914 | 40.19 |
| Grade 1 | 2074 | 21.30 |
| Grade 2 | 785 | 8.06 |
| Not classified/incomplete | 2966 | 30.45 |

years (-0.91) showed decreasing trends, while the other groups (30–59 years and ≥60 years) presented increasing trends.

## Discussion

The present study aimed to investigate the epidemiological situation of leprosy, its trend over the years, and whether there were trends, both in the detection of the disease and in disabilities, analyzing disparities and/or differences regarding gender and age in a hyperendemic metropolis in the Central-West region of Brazil.

The study showed that leprosy has been declining in the research area; however, when analyzed according to gender, age and disabilities, we observed that the leprosy affects men, children under 15 years and elderly people unequally, with an increase in disabilities, raising the

**Detection total**

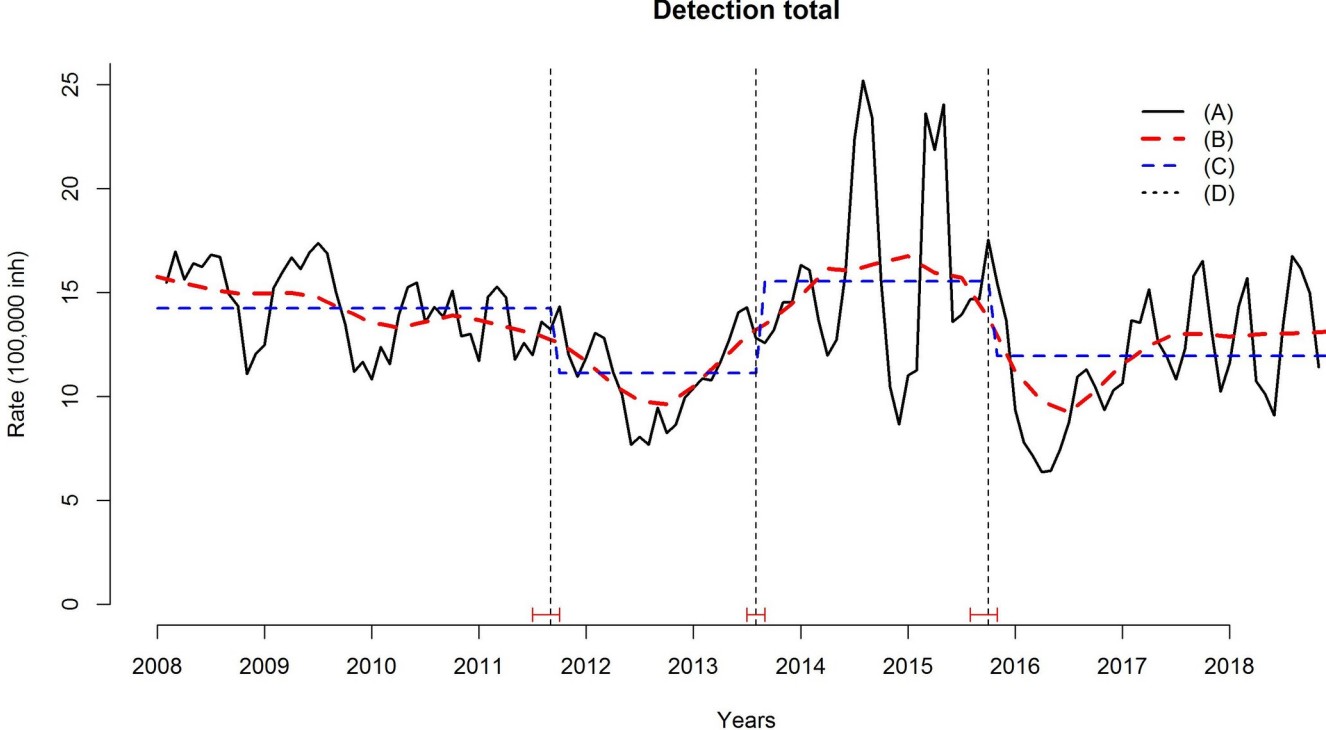

**Fig 2. Changes in the structure of the time series for the general leprosy detection rate, Cuiabá, Mato Grosso, Brazil (2008–2018).** (A) Time series; (B) Trend; (C) Change of structure; (D) Point of structural change.

hypothesis that late diagnosis and underreporting may be occurring, revealing a possible weakness of the health services in Cuiabá [31–33].

A current ally to fight leprosy is leprosy chemoprophylaxis. According to the WHO guidelines, single-dose rifampicin (SDR) as post-exposure prophylaxis (PEP) can be used in children and adults [1]. In a randomized controlled trial, SDR given to leprosy contacts provided a reduction in leprosy risk of 57% in 2 years and 30% in 5–6 years [34]. As leprosy is a highly stigmatized disease, revealing the identity of the index patient when implementing this preventive therapy for contacts should be handled with care and only after gaining consent, especially when this takes place outside the patient's family.

The result that men are more affected than women has also been found in other studies [35,36]. This can be related to several factors, such as being less concerned about their own health and difficulties for men to access public health services [37–39]. Currently, there are few health policies aimed at this population to meet their needs [40]. Barriers related to the difficulty of access to health services, the incompatibility between the hours of operation of health units and the workday, and the belief of being less susceptible to the disease in comparison with women may contribute to this greater burden of leprosy in the male population [41].

Most of the cases had incomplete elementary education, an indicator of low schooling, which may be related to the social aspect and living conditions. Low levels of education hinder access to better jobs and better economic conditions [42–43].

The predominance of multibacillary cases, with the most severe clinical forms (especially borderline and lepromatous cases), may suggest the occurrence of active transmission of the disease and, consequently, greater potential to incapacitate the affected individuals [44].

**Table 3. Mean percentage variation in the rates of detection of leprosy and disability grade at the time of diagnosis of cases, in an endemic municipality in Central-West Brazil (2008–2018).**

| Variables | Average Monthly Percentage Change (AMPC) (%) | Trend* |
|---|---|---|
| **LEPROSY DETECTION** | | |
| General population | - 0.11 | Decreasing |
| **Gender** | | |
| Male | 0.01 | Increasing |
| Female | - 0.26 | Decreasing |
| **Age group (years)** | | |
| <15 | -0.99 | Decreasing |
| 15–29 | -0.49 | Decreasing |
| 30–59 | 0.07 | Increasing |
| ≥60 | -0.0028 | Decreasing |
| **DISABILITIES** | | |
| **Disability grade in patients with leprosy** | | |
| Grade 0 | -0.57 | Decreasing |
| Grade 1 | 0.56 | Increasing |
| Grade 2 | 0.38 | Increasing |
| Not evaluated | 0.28 | Increasing |
| **Disability grade in male patients** | | |
| Grade 0 | -0.44 | Decreasing |
| Grade 1 | 0.35 | Increasing |
| Grade 2 | 0.67 | Increasing |
| Not evaluated | 0.22 | Increasing |
| **Disability grade in female patients** | | |
| Grade 0 | -0.7 | Decreasing |
| Grade 1 | 1.24 | Increasing |
| Grade 2 | -6.09 | Decreasing |
| Not evaluated | 0.45 | Increasing |
| **Disability grade in children <15 with leprosy** | | |
| Grade 0 | -0.26 | Decreasing |
| Grade 1 | 1.02 | Increasing |
| Grade 2 | 9.86 | Increasing |
| Not evaluated | -0.42 | Decreasing |
| **Disability grade in patients aged 15 to 29 years** | | |
| Grade 0 | -0.56 | Decreasing |
| Grade 1 | 0.98 | Increasing |
| Grade 2 | 0.04 | Increasing |
| Not evaluated | -0.91 | Decreasing |
| **Disability grade in patients aged 30 to 59 years** | | |
| Grade 0 | -0.47 | Decreasing |
| Grade 1 | 0.74 | Increasing |
| Grade 2 | -0.62 | Decreasing |
| Not evaluated | 0.63 | Increasing |
| **Disability grade in patients aged ≥60 years** | | |
| Grade 0 | -0.40 | Decreasing |
| Grade 1 | 0.73 | Increasing |
| Grade 2 | 0.67 | Increasing |

(*Continued*)

**Table 3.** (Continued)

| Variables | Average Monthly Percentage Change (AMPC) (%) | Trend* |
|---|---|---|
| Not evaluated | 0.21 | Increasing |

* Considered the mean, which is influenced by changes and/or extreme variations.
Source: authors.

In relation to children under 15 years of age, when analyzing the disability grade, there were growing trends in the number of G1D and G2D in this age group, which may indicate that the municipality faces difficulties in the early diagnosis of the disease and ongoing transmission. A study by Xavier et al. (2014) with children under 15 years of age indicated that the early exposure to the pathogen in this age group suggests a late diagnosis and prolonged exposure [45]. The fact that there are any G2D in children is concerning, as it falls far short of the WHO goal for zero disability in children [1,45]. In addition, people who are affected by multibacillary forms of the disease have a greater chance of developing health problems. . . Leprosy is highly disabling when not properly treated in this population, which can influence academic school performance (and future occupation) and cause problems related to social limitations, discrimination, self-esteem, and stigma experienced by the affected person, especially because this is a period of growth and physical and emotional development [45,46].

In the state of Mato Grosso in 2015, the National Campaign for Leprosy, Geohelminthiasis and Trachoma ["Campanha Nacional de Hanseníase, Verminoses, Tracoma e Esquistossomose"] was initiated, which mobilized local health services to execute actions related to the active search for cases, focusing on schoolchildren aged 5 to 14 years. The campaign was carried out in approximately 915 schools in 65 municipalities (including Cuiabá), to examine and treat more than 291.200 students and their possible contacts [21]. It is estimated that the campaign may have had an impact in the region studied, reflecting the peak of detection verified in the study for this age group. No further policies were encountered in public archives to influence the detection of leprosy in the region during the time period.

Regarding the age group of 15 to 29 years, despite presenting a decreasing trend in the case detection rate, it should be noted that G1D and G2D ended the series with increasing trends. These results can be strong indications of late diagnosis and the existence of underreported of cases [42,47], since the decrease in the detection rate of new cases is not accompanied by the decrease of cases with disability grade. This constitutes a warning about possible difficulties of health services in detecting patients early and conducting adequate active case finding activities [48].

The group aged 30 to 59 years ended the time series with an increasing trend in the detection rate of new cases, as well as in G1D. This age group is composed of the economically active population, where the disabilities and incapacities caused by leprosy affect the work and social life environments, causing not only economic losses for the individual and his or her community but also psychological losses [9,49,50].

The older adult age group (aged 60 years or more) had a decreasing trend in the rate of detection of new cases and increasing values in G1D and G2D. As stated, the antagonism between the decrease in the detection rate and the increase in disabilities is a strong indication of difficulties in the active search for cases and the possibility of underreporting [47].

The issue of not evaluating cases for disability showed an increasing trend over the study period. The assessment of physical disability at the time of diagnosis is a priority action for newly diagnosed leprosy cases, and the fact that 'non-evaluation' presents an increasing trend may raise discussions regarding the management of leprosy cases in the research setting.

According to the Ministry of Health quality control, at least 75.00% of new cases at the time of diagnosis need to be evaluated for the disability grade and registered into SINAN [23,24]. In the Official Epidemiologic Bulletin of Cuiabá of the years 2017, 2018 and 2019 health units evaluated a mean of 56.24% G2D registration of new cases at the time of diagnosis, below expectations [22]. The authors of this study emphasize that a 100% of disability grade assessment and registration at time of diagnosis, as well as for disease progression evaluation, should be pursued.

These findings lead us to suppose that the assessment and registration of leprosy patients at health care level or in the national surveillance system had not been carried out systematically, with the number of unevaluated cases often showing a lack of adherence, unpreparedness, and a lack of standardized protocols that guide the classification and registration, of both the clinical leprosy form and its disability grade.

Leprosy affects men and women, age groups and social classes in different ways, it emerges from inequality and produces more inequality among the affected populations. In addition to this discussion, it can also be defined as neglected, both by public policies and health authorities, with regards to incorporating it into priority investment actions, considering that it affects the most marginalized populations and those in situations of extreme vulnerability [51–53].

If there is a clear intention to overcome leprosy, investments in health care and different methodological approaches must be considered, in both research, health service delivery and the surveillance system of health systems and territories. The elimination of leprosy involves comprehending the differences in gender issues and life stages in the health territories, and the elaboration of intervention projects aimed to target risk groups must be sufficiently supported by scientific and operational evidence.

Guiding public policies based on this evidence is essential for advancing equity and eliminating leprosy. Actions are needed to support health care providers to correctly evaluate disabilities. Active case finding activities can result into higher patient detection rates, and in addition, the search and follow-up of patient contacts must also be promoted. As mentioned, SDR-PEP could be studied by policy makers and/or health care managers, to be implemented in their context based on the WHO's recommendations and the effectiveness and feasibility as reported in the literature [1,7, 34,54,55].

A limitation of this study is that the database used is secondary, so it may contain inconsistent information regarding quantity and quality with presence of data that were potentially ignored or incomplete, data regarding the operational classification and clinical leprosy registration form were, in some cases, not filled completely. Another limitation involved STL, which is only a visual resource for showing findings and does not having a "measure" of increase or decrease. AMPC is based on a percentage, which is why it does not contain a p-value or 95% confidence interval. In the data, we encountered some operational classification with incompatibility with clinical form, such as 'indeterminate' being classified as 'multibacillary' cases, thus having to add 'not classified/incomplete' on operational classification. Not all clinical forms were filled in the databank contrary to the operational classification, causing disparities. The number of children with G2D in the trends is small and thus might not be able to show a clear trend. The interpretation of these data cannot be understood at the individuals' level.

In conclusion, the present study highlights the need to prioritize leprosy active case finding activities to foster early detection, to improve the care for this disease, as well as to develop strategies for leprosy prevention and health care strengthening.

## Supporting information

**S1 File. STROBE Statement—checklist of items that should be included in reports of observational studies.**
(DOCX)

**S1 Dataset. Minimal anonymized data set.**
(XLSX)

**S1 Fig. Trends in total detection rates, in the general population, by gender and age groups per 100.000 inhabitants in Cuiabá (2008–2018).** (Black line) Time series; (Red line) Trend.
(TIF)

**S2 Fig. Trends in the gross number of cases of disability grade at Diagnosis (DPD) in patients diagnosed with leprosy in the period from 2008 to 2018 in Cuiabá.** (Black line) Time series; (Red line) Trend.
(TIF)

**S3 Fig. Trends in the gross number of cases of disability grade at Diagnosis (DPD) by gender in patients diagnosed with leprosy in the period from 2008 to 2018 in Cuiabá.** (Black line) Time series; (Red line) Trend.
(TIF)

**S4 Fig. Trends in the gross number of cases of disability grade at Diagnosis (DPD) by age group in patients diagnosed with leprosy in the period from 2008 to 2018 in Cuiabá.** (Black line) Time series; (Red line) Trend.
(TIF)

## Acknowledgments

The authors would like to thank the Health Surveillance Service of the Cuiabá Regional Health Management Unit of the state government of Mato Grosso for making the data available.

## Author Contributions

**Conceptualization:** José Francisco Martoreli Júnior, Antônio Carlos Vieira Ramos, Ricardo Alexandre Arcêncio.

**Data curation:** José Francisco Martoreli Júnior, Antônio Carlos Vieira Ramos, Josilene Dalia Alves.

**Formal analysis:** José Francisco Martoreli Júnior, Antônio Carlos Vieira Ramos, Ricardo Alexandre Arcêncio.

**Investigation:** José Francisco Martoreli Júnior.

**Methodology:** José Francisco Martoreli Júnior, Antônio Carlos Vieira Ramos, Thaís Zamboni Berra, Dulce Gomes, Carla Nunes, Ricardo Alexandre Arcêncio.

**Project administration:** José Francisco Martoreli Júnior, Antônio Carlos Vieira Ramos.

**Resources:** Antônio Carlos Vieira Ramos.

**Supervision:** Antônio Carlos Vieira Ramos, Ricardo Alexandre Arcêncio.

**Visualization:** José Francisco Martoreli Júnior, Antônio Carlos Vieira Ramos, Josilene Dalia Alves, Juliane de Almeida Crispim, Luana Seles Alves, Thaís Zamboni Berra, Tatiana Pestana Barbosa, Fernanda Bruzadelli Paulino da Costa, Yan Mathias Alves, Márcio Souza dos

Santos, Dulce Gomes, Mellina Yamamura, Ione Carvalho Pinto, Miguel Angel Fuentealba-Torres, Carla Nunes, Flavia Meneguetti Pieri, Marcos Augusto Moraes Arcoverde, Felipe Lima dos Santos, Ricardo Alexandre Arcêncio.

**Writing – original draft:** José Francisco Martoreli Júnior, Antônio Carlos Vieira Ramos, Ricardo Alexandre Arcêncio.

**Writing – review & editing:** José Francisco Martoreli Júnior, Antônio Carlos Vieira Ramos, Josilene Dalia Alves, Juliane de Almeida Crispim, Luana Seles Alves, Thaís Zamboni Berra, Tatiana Pestana Barbosa, Fernanda Bruzadelli Paulino da Costa, Yan Mathias Alves, Márcio Souza dos Santos, Dulce Gomes, Mellina Yamamura, Ione Carvalho Pinto, Miguel Angel Fuentealba-Torres, Carla Nunes, Flavia Meneguetti Pieri, Marcos Augusto Moraes Arcoverde, Felipe Lima dos Santos, Ricardo Alexandre Arcêncio.

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
