## [Decision Letter · Decision Letter 0]

15 Apr 2021

Dear Mr Martoreli Júnior,

Thank you very much for submitting your manuscript "Inequality of gender, age and
disabilities due to leprosy and trend in a hyperendemic metropolis: Evidence from an
eleven-year time series study in Central-West Brazil" for consideration at PLOS
Neglected Tropical Diseases. As with all papers reviewed by the journal, your
manuscript was reviewed by members of the editorial board and by several independent
reviewers. In light of the reviews (below this email), we would like to invite the
resubmission of a significantly-revised version that takes into account the
reviewers' comments. 

We cannot make any decision about publication until we have seen the revised
manuscript and your response to the reviewers' comments. Your revised manuscript is
also likely to be sent to reviewers for further evaluation.

Sincerely,

Alberto Novaes Ramos Jr

Associate Editor

Hélène Carabin

Deputy Editor

Reviewer's Responses to Questions

**Key Review Criteria Required for Acceptance?**

**Methods**

-Are the objectives of the study clearly articulated with a clear testable hypothesis
stated?

-Is the study design appropriate to address the stated objectives?

-Is the population clearly described and appropriate for the hypothesis being
tested?

-Is the sample size sufficient to ensure adequate power to address the hypothesis
being tested?

-Were correct statistical analysis used to support conclusions?

-Are there concerns about ethical or regulatory requirements being met?

Reviewer #1: methods are well explained and appropriate for this study

Reviewer #2: Information on the objectives of the study is clear, ethical approval
was obtained. See full comments below in the Methods section.

Reviewer #3: The objectives of the study are clearly articulated with a clear
testable hypothesis stated.

The study design is appropriate to address the stated objectives.

The population is clearly described and appropriate for the hypothesis being
tested.

Since all the population is included in the study, there is no sample size issue.

The statistical analysis is corrected for supporting the conclusions.

But: (1) Please include the diagnosis criterion of leprosy for better under standing.
(2) Please clarify the mobile population for analysis since it could be a
confounding factor for the result. (3) Is there any policy change for leprosy which
could influence the detection of leprosy. (4) The method of Seasonal-Trend
decomposition procedure based on Loess (STL) can be described more clearly for
better understanding by readers.

Reviewer #4: This is an interesting exercise made possible by the comprehensive SINAN
database in Brazil. The authors have given much emphasis to the statistical analysis
of trends. I believe the statistical approach is generally valid and the analysis
well-performed, but there are some unclarities:

1) Please provide the definition used in Brazil for hyper endemic. For international
use, consider to just state 'highly endemic' and use this term as well in the title,
instead of 'hyper endemic'.

2) There is quite some 'fog' in relation to disability grade, its registration and
the way it was dealt with in the analysis. Lines 152-154 are unclear; e.g. what does
'below expectations' mean. Also the sentence in lines 188-190 is unclear. What total
number? And how does this translate in the results? There I see a breakdown by
disability grade and 'not registered'. With the great uncertainty with regard to the
indicator disability grade, why do you not just take 'grade 0' and 'joint grade 1
and 2' in the trend analysis, and ignore the 'not evaluated'?

3) I think very important information is missing necessary for interpreting trends,
namely operational changes in the leprosy control program, or for that matter the
health system in general. The total leprosy detection over the ten years is rather
spiky. It comes down between 2008 and 2012, increases and decreases steeply between
2012 and 2016, and then increases again to reach possibly a plateau. The health
system (or other operational changes) during this time need to be described in some
detail in the methods section under a separate heading. This is essential context to
understand and interprete the trends.

Reviewer #5: There is too much background information in the methods – make this more
concise, especially the parts about the “study design and research scenario”. Also,
in English, we generally don’t say “Research scenario”. I would say “Study Design
and Setting”.

**Results**

-Does the analysis presented match the analysis plan?

-Are the results clearly and completely presented?

-Are the figures (Tables, Images) of sufficient quality for clarity?

Reviewer #1: well presented and relevant information provided

Reviewer #2: The result section is clearly written. I would suggest to use not only
colours, but also a different line appearance in the table to make them more clear
for readers who are printing the tables in black/white. Please also see my comments
below on the Results section.

Reviewer #3: The analysis presented matches the analysis plan.

The results are clearly and completely presented.

But: A standardized population need to be provided and calulated for the age and
gender groups.

Reviewer #4: With regard to the case characteristics (table 1) there is some
unclarity about operational classification and clinical form. PB (26.67%) seems to
be the sum of I and TT clinical forms (26.88%), but not just the same. In the
sentence starting on line 237 it states: "... there was a predominance of
multibacillary cases (73.33%), followed by borderline cases (57.00%)". But
borderline cases are part of MB... 

With regard to presentation of trends, I would first like to see the figure
'Detection total' separately with under the time line some bars or arrows indicating
important operational changes in time. See for instance the figure A in the
following article for inspiration https://www.cell.com/trends/molecular-medicine/fulltext/S1471-4914(20)30065-4.
In that way the overall trend is placed in context of the most important operational
factors influencing the trend. 

Table 2 summarizes the main findings of the trend figures 2-5. You could therefore
consider to place all these figures in an appendix.

In table 2, make sure to give the full name of the abbreviation AMPC. Furthermore, as
reader I need help to understand the importance of the figures under AMPC. Under the
heading Trend you state Increasing or Decreasing. But is there a way to indicate a
measure of increase or decrease, for instance through a p-value? And how do you
establish the importance of a trend. When is the trend such, that you want to
highlight it as an important result and discuss its implications. And when is it
just some irrelevant 'noise'.

Reviewer #5: 1. I don’t think G0D (as opposed to G1D and G2D for grade 1 and grade2)
is an accepted acronym for “no disability” and it’s confusing to say that G0D is
decreasing – G0D in effect is “NO disability”. I would use a different way of
describing it because if you read it quickly you may think that some disability is
decreasing whereas what you are saying is that people presenting without disability
is actually decreasing among many groups which is very concerning. 

For all “G0D”, I would substitute “No disability”

2. I know that Asian descent for race translates to “amarelo” in Portuguese, but many
English language readers would find “Yellow” offensive, so I would change that to
Asian descent. 

3. Lines 237-238: You wouldn’t really say “followed by borderline cases”, since they
are part of two different categories, just state these separately.

Figures overall seem ok, but would change the G0D label as per above.

**Conclusions**

-Are the conclusions supported by the data presented?

-Are the limitations of analysis clearly described?

-Do the authors discuss how these data can be helpful to advance our understanding of
the topic under study?

-Is public health relevance addressed?

Reviewer #2: In the discussion, I am missing the role of leprosy prevention as
post-exposure chemoprophylaxis with a single dose of rifampicin (SDR-PEP) for
contacts of leprosy contacts, in combination with contact screening / active case
finding. This is a very important strategy for decreasing the number of new cases /
case detection delay / disability etc. For example in Brazil, a current research
project is ongoing studying advanced chemoprophylaxis regimen (PEP++). But the
evidence from e.g. COLEP & LPEP regarding SDR-PEP is clear and it is also
included in the WHO Leprosy Guidelines for the Diagnosis, Treatment and Prevention
of Leprosy. You could add this as reconmendation. 

Including 1 study limitation may be a bit limited.

Reviewer #3: The conclusions are supported by the data presented.

The limitations of the study are addressed.

Public health relevance is addressed.

The authors need to discuss in more detail to help readers understanding the
result.

Reviewer #4: To start with, cut the discussion by at least 50%. It is far too long
and therefore unclear. In the discussion state first what you consider the really
most important findings of your study. Which trends are really relevant in terms of
being unexpected, undesired and amendable to health care interventions?

After stating the most important findings, only then start with some explanation and
interpretation. Make clear that you are discussing everything against the Brazilian
background. You use the term 'global' at a certain point, but I understand it to
mean Brazil as a whole and not the world. Here it is also important to take into
account the operational factors, that you have now incorporated in the new figure
1.

Much of the disparities are due to delayed detection, which may apply more strongly
to e.g. women and people in the working age. This is also related to stigma. Try to
bring that together in the discussion more comprehensively, instead of just going
along the different groups (gender, age and disability) in order of appearance in
analysis.

Whereas the general discussion text should be much shorter, the paragraph on
strengths and weaknesses should be a more detailed.

Finally, the conclusion is very general. Can you make it a bit more specific, based
on your findings? Just one or two sentences more.

Reviewer #5: 1. Line 288 Again, do not say “grade 0 disabilities” are decreasing.
That’s misleading. Grade 0 means there is no disability, so you shouldn’t describe
it like that. You should say that the rates of new diagnosis without disability is
decreasing. 

2. Gender differences – I think you are missing discussion of a potentially big
reason for differences in gender that has been proposed – that women are not seeking
care for their leprosy, possibly due to more PB in women than men so
underrecognition or possibly due to stigma (reference: Factors preventing early case
detection for women affected by leprosy: a review of the literature. Price VG.Glob
Health Action. 2017 Jan-Dec;10(sup2):1360550. doi:
10.1080/16549716.2017.1360550.PMID: 28853325 )

3. The discussion is too long – you should summarize the different age time series
more succinctly and in 1-2 paragraphs and not just repeat your results. 

4. There is no discussion on the limitations of this study – one big thing, which you
can tell from the figures, is that the number of children with grade 2 disability is
actually quite small so it may not be able to tell a clear trend. However, the fact
that there are any G2D in children is concerning, and at odds with the WHO goals for
zero disability in children < 15 (you should mention these goals in the
discussion).

**Editorial and Data Presentation Modifications?**

Reviewer #2: ABSTRACT:

32-35: First sentence in abstract is quite long, change this into 2 sentences to
increase readability.

35-36: Second sentence in abstract is incorrect. Start the sentence with e.g. “This
is a …”

45-46: In the abstract it is stated “Regarding detection according to gender, there
was a decrease among women”, but “by evidencing a increasing trend of leprosy cases
among women” (47-48). This seems contradictive, please make some changes. It should
also be ”aN increasing trend”.

INTRODUCTION:

60: In introduction, I would advise to change “which lead to physical disabilities”
into “which can lead to physical disabilities”. 

61: Please remove “skin contact” in “skin contact or other means cannot be excluded”,
the evidence for skin contact is low (i.e. using a tattoo needle in a leprosy patch
and next tattooing another person with the same needle is risky, but not normal skin
contact and rare of course) and stating it like this can sound stigmatizing for some
people. 

65: Reference numbers should be placed before/after (depending on reference style)
punctuation marks. In this case: before a comma (,) or full stop/period (.), not in
the middle of a sentence. Please move [2,3] to after “imagined”.

67: “In 2016, according to WGHO data, 143 countries…” -> This data is a bit
outdates, especially because your data reflected in the time series study is ranging
from 2008-2018. Please use WHO data from 2018 (or even 2019) here. 

72: It seems like you are referring to both North and South America, so it should be
continentS.

77: “preventing cases with grade of disability” should be “preventing cases with
disabilities”.

79: Please remove the capital at “The Plan” -> “The plan” or even better, change
it to “The strategy”.

89: Place a comma after “2010-2012”.

102-107: This is also the case because you cannot count ‘undetected cases’.

105: “disabilitie” should be “disability”.

111: the word “public” is missing: “define public policies” -> “define public
health policies”.

113: the word “tools” is missing: “more sensitive is the time series” -> “more
sensitive tools is the time series”.

METHODS

129: when including such an exact number as 3,266,538 km2, I would suggest to remove
the word “approximately”.

138: Explain shortly what Gini index is.

140. full stop is missing after “[17,18]”

140-141: capital usage is incorrect, please check. I would suggest to remove most of
the capitals used in this sentence.

145-146: same point regarding the capitals, I would suggest to remove most of the
capitals used in this sentence.

153-154: which expectations are meant here? Where does 75.00% come from?

STUDY POPULATION AND INFORMATION SOURCES

164: I do understand why you included this (race/skin color), and the explanation on
this in the discussion is very clear and correct. But when people are reading this
in the Methods section, it is a bit in your face. Registering this data should be
avoided or happen very carefully from an ethical perspective, as it may sound like
ethical profiling / offensive / discriminatory to some people. The colors as
outlined here (“white, black, yellow, mixed, and indigenous”) is not ethical
preferred. Starting with white is tricky (also now with Black Lives Matter),
alphabetical order would be better. I would include information that people had to
register their own race (instead of the health professionals / researchers
registering it, because it is hard to judge whether someone is mixed or black for
example). I would remove it completely from the article. If you want to leave it in,
I would suggest to more clearly explain also in the Methods section that this is
registered in the national leprosy system and that this is outdated or not found to
be ethical anymore and why this data can still be important and change/remove this
listing of skin tones. 

181/189/192/212/214/215: Grade of Disability is sometimes written with capitals and
sometimes without, I would suggest to write it without capitals. I also prefer
“disability grade” over “grade of disability”.

202-206: I would use “” or ‘’ for the words you are defining: ‘Trend’, ‘Seasonality’,
‘noise’.

RESULTS:

229 & Table & 321: see comments above about “race/skin color”

235 & 262: Delete “Source: Authors”. If there is no other source included, it is
clear it is from the authors.

238/240/241-242/245/Table 2/260/262/264/268/268/271/274/354/372: See comment above:
“Grade of Disability” -> “disability grade”.

243: full stop missing after “Fig”.

 Table 2: “General Population”-> “General population”.

Fig. 2/3/4/5: It may be nice to use strips/dots for one of the lines in the graphs
(e.g. the red one) besides a color difference, so people who print it black &
white can still see the color difference.

279 & 339 & 434: I would suggest a different word here, as “behavior“ is more
suitable for humans/animals, go for e.g. “trend” here instead.

280 & 289: “and or” should be “and/or”.

297: remove “the” before “: disability”.

299: Remove “The” before “literature”.

300: Maybe add reference here, as “literature” sounds like plural.

308: Possibly remove “various”.

310: you could replace “published works” by simply staying “studies”.

311: what is meant here by “risky situations”? It this referring to getting infected?
As prolonged contact is needed, I would not call this “risky situations” (also
sounds a bit stigmatizing), or is it referring to possible damage to the hands
because of work related accidents or so? I would delete “risky situations” or
rephrase and explain it.

313-314: at least 2 references seem to be missing, as you are talking about evidence
from the 90ies and current policies.

343: add the word “the” between “regarding” and “detection”.

343 & 354: add “years of age” after “15”.

348-349: word repetition “carry out”, if you like, change for executed/organized…

357 & 377: reference location not correct (should be at end of sentence or before
comma).

357-360: what do you mean here by the statement that “early exposure” leads to
disabilities? Because of the length of the exposure/infection period (although, they
were still children at time of diagnosis, so it cannot be decades for example) or is
it because their bodies were still developing biologically at that age. Or both?

362: I would change “academic performance” into “school performance”, as “academic”
often refers to university level.

383: delete commas around “as well as”.

384: “incapacity” –> “incapacities”.

400: change the word “scenario” into “setting”.

400: you may like to add “registration at health care level/national surveillance
system”.

402: you may like to add “lack of adherence”.

408: “a way of” sounds a bit informal, you could replace it for “a method of”.

421-422: change “does not behave in a homogenous way” into “disease trend is not
homogenous”.

427: what is meant by “promotion actions”? Awareness raising? Self care? Please
explain.

431 & 432: I don’t think “State” should be capitalized here.

432: “State protection” has a double definition (also a negative one), I would
replace it for “state support”.

434: 432-435: references are missing, as you are referring to multiple studies and
WHO reports.

435: delete “which is true”, as you need to have verified all these studies/data to
know for sure.

435: a decrease in new cases can also be caused by a lack of active case finding
activities, and this, is not always a ‘positive’ finding.

445: usually, multiple limitations (at least 2) are named.

Say something on post-exposure prophylaxis (PEP) as prevention strategy for leprosy
(see my comment above) in the discussion section. 

REFERENCES:

Reference 1 (457) and reference 5 are the same, though 2 difference websites are
used. Please replace reference 1 with number 5 (2018 WHO is the correct reference
for the WHO Leprosy Guidelines).

Reviewer #4: The first paragraph of the introduction is rather outdated. Please use
2019 WHO data and also refer to the latest WHO strategic document for leprosy.

Reviewer #5: The paper needs some work before publication, especially in its length,
which I think can be shortened, and in the English / writing in certain places. I
found a few typos as well. It needs a thorough review for English language editing,
spelling, typos before publication. I also didn't see cover letter or summary which
was unusual.

PLOS authors have the option to publish the peer review history of their article
(what does this mean?). If published, this will
include your full peer review and any attached files.

If you choose “no”, your identity will remain anonymous but your review may still be
made public.

**Do you want your identity to be public for this peer review?** For
information about this choice, including consent withdrawal, please see our
Privacy Policy.

Reviewer #1: Yes: Carlos Franco-Paredes

Reviewer #2: No

Reviewer #3: No

Reviewer #4: No

Reviewer #5: No

**Summary and General Comments**

Reviewer #1: I enjoyed reading this and reviewing this manuscript. It is a master
piece in social sciences in medicine and it reveals the fallacy of the academic
imperlalism by the "experts" drinking latte in European countries dictaing polices
in coutnries where leprosy is a major public health concern. This publication is a
clear demonstration of the poor decision meaking of public health organizations. The
only thing that the elimination campign of leprosy did, was the elimination of
attention to major chronic infectious disease that causes severe disability. It was
an honor to review this paper.

Reviewer #2: Very nice article and an important topic. Please make some editorial
changes (see full comments), and pay extra attention to capital usage. Also, be
careful with the section on race/skin color (especially in the Methods section) as
this is a sensitive topic. See if you can add another limitation and add
chemoprophylaxis for leprosy in your recommendations. Also please do check the
reference list again and add a few extra references in the text (see comments).

Reviewer #5: This study is a straightforward analysis of epidemiologic data on
leprosy from the SINAN data base and uses a time series analysis that really studies
the trends over time of the 3 key epidemiologic indicators for leprosy: 1. New case
detection rate, 2. Pediatric cases, 3. New grade 2 disability. Using a hyperendemic
city is a really nice way to not only show concerning trends in this area, but to
model ways that we can study other geographic areas. These findings are very
important because it shows that the decreasing incidence does not tell the whole
study and that different age / sex can be impacted differently and may need
different control strategies. The paper needs some work before publication,
especially in its length, which I think can be shortened, and in the English /
writing in certain places.
---

## [Decision Letter · Decision Letter 1]

9 Jul 2021

Dear Mr Martoreli Júnior,

Thank you very much for submitting your manuscript "Inequality of gender, age and
disabilities due to leprosy and trends in a hyperendemic metropolis: Evidence from
an eleven-year time series study in Central-West Brazil" for consideration at PLOS
Neglected Tropical Diseases. As with all papers reviewed by the journal, your
manuscript was reviewed by members of the editorial board and by several independent
reviewers. The reviewers appreciated the attention to an important topic. Based on
the reviews, we are likely to accept this manuscript for publication, providing that
you modify the manuscript according to the review recommendations. Please pay
particular attention to reviewer's 2 comments and make sure that their concerns are
addressed appropriately.

Sincerely,

Alberto Novaes Ramos Jr

Associate Editor

Hélène Carabin

Deputy Editor

Reviewer's Responses to Questions

**Key Review Criteria Required for Acceptance?**

**Methods**

-Are the objectives of the study clearly articulated with a clear testable hypothesis
stated?

-Is the study design appropriate to address the stated objectives?

-Is the population clearly described and appropriate for the hypothesis being
tested?

-Is the sample size sufficient to ensure adequate power to address the hypothesis
being tested?

-Were correct statistical analysis used to support conclusions?

-Are there concerns about ethical or regulatory requirements being met?

Reviewer #1: OK

Reviewer #2: TITLE: 

- "hyperendemic" is written with space in the rest of the manuscript, but without in
the title

INTRODUCTION:

- Most changes made were appreciated.

- See my suggested changes and comments in the text file attached. 

- Please include the latest version oif the WHO Global Leprosy Strategy
(2021-2030).

- I would delete: "Also, regarding the inequality related to age, it is known that
when there is a delay in diagnosis, children that who had contact with index cases
can also become ill [1112], which is an important gap to be filled."

METHODS:

- Most changes made were appreciated.

- See my suggested changes and comments in the text file attached. 

- You could think of adding a definition table with an overview of the disability
grades (what is G0D, G1D, G2D)

- Capital usage still not fully correct

- This sentence is unclear: "Regarding the disability gradethe grade of disability,
health units evaluated a mean of 56.24% of new cases with a disability at the time
of diagnosis, below expectations, which was listsed as should be at least 75.00%
according to SINAN [235, 26]." 

- I donnot understand this sentence because of the level of English, you can maybe
also delete it: "For the construction of time series of the cases with disability
grade we considered all the number of cases with disability grade (G0D, G1D, G2D,
and not evaluated)."

- This is also quite vague, could it be deleted? "... even for a very long time
series and large amounts of trend and seasonal smoothing, a very small amount of
trend smoothing, and seasonal components that are not distorted by aberrant behavior
in the data ability to decompose time series with missing values"

- This should not be in the methods section, but in the discussion (is already
included there). Also, the word 'dubious' is too strong/incorrect: "Finally, we
mention that the notification forms are filled out by third parties, so dubious data
may be provided."

Reviewer #3: The objectives of the study are clearly articulated with a clear
testable hypothesis stated. The study is design appropriate to address the stated
objectives. The population is clearly described and appropriate for the hypothesis
being tested. The sample size is sufficient to ensure adequate power to address the
hypothesis being tested.

Reviewer #4: See attachment

**Results**

-Does the analysis presented match the analysis plan?

-Are the results clearly and completely presented?

-Are the figures (Tables, Images) of sufficient quality for clarity?

Reviewer #1: OK

Reviewer #2: - Changes made were appreciated.

- 'ignored' is too strong in table 1, go for: 'not classified / incorrect /
incomplete' instead

Reviewer #3: The analysis presented matches the analysis plan. The results are
clearly and completely presented.

Reviewer #4: See attachment

**Conclusions**

-Are the conclusions supported by the data presented?

-Are the limitations of analysis clearly described?

-Do the authors discuss how these data can be helpful to advance our understanding of
the topic under study?

-Is public health relevance addressed?

Reviewer #1: OK

Reviewer #2: DISCUSSION:

- Most changes made were appreciated.

- See my suggested changes and comments in the text file attached. 

- Improvements can be made regarding: item order in the discussion, repetition, level
of English.

- I would suggest to delete this, as it seems less relevant (not clearly explained
how this affects the leprosy programme) and the discussion word count is too high:
"In the state of Mato Grosso, in 2015, the “National Campaign for Leprosy,
Geohelminthiasis and Trachoma” was initiated, which mobilized local health services
to execute carry out actions related to the active search for cases, focusing on
schoolchildren, aged from 5 to 14 years. The campaign was carried out in
approximately 915 schools in 65 municipalities (including Cuiabá), to examine and
treat more than 291..200 thousand students and possibly their possible contacts
[242]. It is estimated that the campaign may have had an impact in the region
studied, reflecting the peak of detection verified in the study for this age group.
No further policies were encountered in public archives to influence the detection
of leprosy in the region duringh the time period."

CONCLUSION:

- See my suggested changes and comments in the text file attached.

Reviewer #3: The conclusions are supported by the data presented. The limitations of
the study are clearly described.

Reviewer #4: See attachment

**Editorial and Data Presentation Modifications?**

Reviewer #1: OK

Reviewer #2: revision needed

Reviewer #3: (No Response)

Reviewer #4: I have made some edits to the text (attached, visible with track
changes). That is easier for me than repeating everything in this form one by
one.

**Summary and General Comments**

Reviewer #1: OK

Reviewer #2: Improvements were made and are much appreciated. Nevertheless, the
manuscript still need to be revised. Especially the discussion needs attention, the
text order should be changed and repetition should be avoided. Both the introduction
and discussion section can be shorter and more to the point. The newly written text
segments need improvents regarding the level of English.

Reviewer #3: (No Response)

Reviewer #4: The paper has improved well after a thorough revision. There are still
some small issues that I have addressed directly in the revised manuscript with
track changes visible. This is not intended as a full language edit, but just to
improve on some relatively important matters with regard to language and
explanation.

PLOS authors have the option to publish the peer review history of their article
(what does this mean?). If published, this will
include your full peer review and any attached files.

If you choose “no”, your identity will remain anonymous but your review may still be
made public.

**Do you want your identity to be public for this peer review?** For
information about this choice, including consent withdrawal, please see our
Privacy Policy.

Reviewer #1: Yes: Carlos Franco-Paredes MD

Reviewer #2: No

Reviewer #3: No

Reviewer #4: No

Figure Files:

Data Requirements:

Reproducibility:

References

Manuscript with Track Changes_corrJHR.docx
---

## [Decision Letter · Decision Letter 2]

27 Aug 2021

Dear Mr Martoreli Júnior,

Thank you very much for submitting your manuscript "Inequality of gender, age and
disabilities due to leprosy and trends in a hyperendemic metropolis: Evidence from
an eleven-year time series study in Central-West Brazil" for consideration at PLOS
Neglected Tropical Diseases. As with all papers reviewed by the journal, your
manuscript was reviewed by members of the editorial board and by several independent
reviewers. The reviewers appreciated the attention to an important topic. Based on
the reviews, we are likely to accept this manuscript for publication, providing that
you modify the manuscript according to the review recommendations. 

Sincerely,

Alberto Novaes Ramos Jr

Associate Editor

Hélène Carabin

Deputy Editor

Reviewer's Responses to Questions

**Key Review Criteria Required for Acceptance?**

**Methods**

-Are the objectives of the study clearly articulated with a clear testable hypothesis
stated?

-Is the study design appropriate to address the stated objectives?

-Is the population clearly described and appropriate for the hypothesis being
tested?

-Is the sample size sufficient to ensure adequate power to address the hypothesis
being tested?

-Were correct statistical analysis used to support conclusions?

-Are there concerns about ethical or regulatory requirements being met?

Reviewer #1: yes

Reviewer #2: - The changes made after last round were a great imporvement.

- The methods section can still be shortened, see if you can delete some sentences,
this will make it easier to read. 

- I suggested in the Word document to move a text section from the 'Methods' to the
'Discussion'.

- Please look at my other suggested changes (tracked changes & yellow marked) and
comments in the Word file.

Reviewer #4: (No Response)

**Results**

-Does the analysis presented match the analysis plan?

-Are the results clearly and completely presented?

-Are the figures (Tables, Images) of sufficient quality for clarity?

Reviewer #1: yes

Reviewer #2: - The changes made after last round were a great imporvement.

- Please look at my other suggested changes (tracked changes & yellow marked) and
comments in the Word file.

Reviewer #4: (No Response)

**Conclusions**

-Are the conclusions supported by the data presented?

-Are the limitations of analysis clearly described?

-Do the authors discuss how these data can be helpful to advance our understanding of
the topic under study?

-Is public health relevance addressed?

Reviewer #1: yes

Reviewer #2: - The changes made after last round were a great imporvement.

- Please look at my other suggested changes (tracked changes & yellow marked) and
comments in the Word file. These were especially focused on the level or
English.

Reviewer #4: (No Response)

**Editorial and Data Presentation Modifications?**

Reviewer #1: no

Reviewer #2: Minor revision, mainly editorial, English language and references.
Especially the methods section can be shortened. See my suggestions & comments
in the Word file.

Reviewer #4: (No Response)

**Summary and General Comments**

Reviewer #1: yes

Reviewer #2: - The changes made after last round were a great imporvement.

- If possible, further shorten the 'Methods'-section.

- Describe where the raw data can be found for reproducibility reasons (see my
comment/addition in 'Ethics'-section).

- Please look at my other suggested changes (tracked changes & yellow marked) and
comments in the Word file.

- Please check the references, some are incorrect in the text and inconsistent in the
reference list.

Reviewer #4: I am happy with all revisions made.

PLOS authors have the option to publish the peer review history of their article
(what does this mean?). If published, this will
include your full peer review and any attached files.

If you choose “no”, your identity will remain anonymous but your review may still be
made public.

**Do you want your identity to be public for this peer review?** For
information about this choice, including consent withdrawal, please see our
Privacy Policy.

Reviewer #1: Yes: Carlos Franco-Paredes

Reviewer #2: No

Reviewer #4: No

Figure Files:

Data Requirements:

Reproducibility:

References

Article with Changes Highlighted_reviewers comments+changes
v3.docx
---

## [Decision Letter · Decision Letter 3]

21 Oct 2021

Dear Mr Martoreli Júnior,

We are pleased to inform you that your manuscript 'Inequality of gender, age and
disabilities due to leprosy and trends in a hyperendemic metropolis: Evidence from
an eleven-year time series study in Central-West Brazil' has been provisionally
accepted for publication in PLOS Neglected Tropical Diseases. However, Reviewer 2
still had a few more recommendations to improve the manuscript a little more. Please
make sure to address the comments from Reviewer 2 when preparing your mansucript for
publication.

Best regards,

Alberto Novaes Ramos Jr

Associate Editor

Hélène Carabin

Deputy Editor

Make the minor changes indicated by the reviewer:

"The manuscript has improved a lot and is, in my opinion, almost ready to be
published. Some relatively minor final comments:

- Please check punctuation marks and paces use throughout the manuscript. Examples:
remove comma in sentence 51; parenthesis in sentence 143 should be placed after
'referrals' instead of 'rehabilitation'; remove full stop in sentence 317; remove
space in sentence 325 & 397.

- In table 'Table 1 - Disability grade and his characteristics', the word 'claw' at
hands & feet at G2D can be seen as stigmatizing language. Better is
'contractures' or 'claw hand deformity'

- The reference for 'WHO disability grading for leprosy' (part of Table 1) is:
https://apps.who.int/iris/handle/10665/42060

- The level of English could be improved at sentence 291-2: 'care should be taken to
reveal the identity of the index patient when implementing this preventive therapy
in contacts, especially outside the patient's family.'. I would like to propose to
change that into: 'revealing the identity of the index patient when implementing
this preventive therapy for contacts should be handled with care and only after
gaining consent, especially when this takes place outside the patient's family'.

- In sentence 397, the English could be improved by removing 'between the sum of
them'."

Reviewer's Responses to Questions

**Key Review Criteria Required for Acceptance?**

**Methods**

-Are the objectives of the study clearly articulated with a clear testable hypothesis
stated?

-Is the study design appropriate to address the stated objectives?

-Is the population clearly described and appropriate for the hypothesis being
tested?

-Is the sample size sufficient to ensure adequate power to address the hypothesis
being tested?

-Were correct statistical analysis used to support conclusions?

-Are there concerns about ethical or regulatory requirements being met?

Reviewer #1: Adequate

Reviewer #2: (No Response)

Reviewer #3: The objectives of the study were articulated with a clear testable
hypothesis stated. The study design is appropriate to address the stated objectives.
The population is described and appropriate for the hypothesis being tested. The
sample size is sufficient to ensure adequate power to address the hypothesis being
tested. There are concerns about ethical or regulatory requirements being met.

**Results**

-Does the analysis presented match the analysis plan?

-Are the results clearly and completely presented?

-Are the figures (Tables, Images) of sufficient quality for clarity?

Reviewer #1: Presented well

Reviewer #2: (No Response)

Reviewer #3: The analysis presented matches the analysis plan. The results are
clearly and completely presented.

**Conclusions**

-Are the conclusions supported by the data presented?

-Are the limitations of analysis clearly described?

-Do the authors discuss how these data can be helpful to advance our understanding of
the topic under study?

-Is public health relevance addressed?

Reviewer #1: Relevant and concordant to previous sections

Reviewer #2: (No Response)

Reviewer #3: The conclusions are supported by the data presented. The limitations of
analysis are described.

**Editorial and Data Presentation Modifications?**

Reviewer #1: (No Response)

Reviewer #2: (No Response)

Reviewer #3: Accept

**Summary and General Comments**

Reviewer #1: (No Response)

Reviewer #2: The manuscript has improved a lot and is, in my opionion, almost ready
to be published. Some relatively minor final comments:

- Please check punctuation marks and paces use througout the manuscript. Examples:
remove comma in sentence 51; parenthesis in sentence 143 should be placed after
'referrals' instead of 'rehabilitation'; remove full stop in sentence 317; remove
space in sentence 325 & 397.

- In table 'Table 1 - Disability grade and his characteristics', the word 'claw' at
hands & feet at G2D can be seen as stigmatizing language. Better is
'contractures' or 'claw hand deformity'

- The reference for 'WHO disability grading for leprosy' (part of Table 1) is:
https://apps.who.int/iris/handle/10665/42060

- The level of English could be improved at sentence 291-2: 'care should be taken to
reveal the identity of the index patient when implementing this preventive therapy
in contacts, especially outside the patient's family.'. I would like to propose to
change that into: 'revealing the identity of the index patient when implementing
this preventive therapy for contacts should be handeled with care and only after
gaining consent, especially when this takes place outside the patient's family'.

- In sentence 397, the English could be improved by removing 'between the sum of
them'.

Reviewer #3: (No Response)

PLOS authors have the option to publish the peer review history of their article
(what does this mean?). If published, this will
include your full peer review and any attached files.

If you choose “no”, your identity will remain anonymous but your review may still be
made public.

**Do you want your identity to be public for this peer review?** For
information about this choice, including consent withdrawal, please see our
Privacy Policy.

Reviewer #1: No

Reviewer #2: No

Reviewer #3: No

---

## [Editor Report · Acceptance letter]

11 Nov 2021

Dear Mr Martoreli Júnior,

We are delighted to inform you that your manuscript, "Inequality of gender, age and
disabilities due to leprosy and trends in a hyperendemic metropolis: Evidence from
an eleven-year time series study in Central-West Brazil," has been formally accepted
for publication in PLOS Neglected Tropical Diseases.

Best regards,

Shaden Kamhawi

co-Editor-in-Chief

Paul Brindley

co-Editor-in-Chief
